# Productivity of Short-Rotation Poplar Crops: A Case Study in the NE of Romania

Iulian-Constantin Dănilă [1,2,3] , Cristian Mititelu [1] and Ciprian Palaghianu [1,2,*]

1 Forestry Faculty, "Ștefan cel Mare" University of Suceava, Universității Street 13, 720229 Suceava, Romania; iuliandanila@usm.ro (I.-C.D.); mititelucristian@yahoo.com (C.M.)
2 Applied Ecology Laboratory, Forestry Faculty, "Ștefan cel Mare" University of Suceava, Universității Street 13, 720229 Suceava, Romania
3 Forest Biometry Laboratory, Forestry Faculty, "Ștefan cel Mare" University of Suceava, Universității Street 13, 720229 Suceava, Romania
* Correspondence: cpalaghianu@usv.ro

**Abstract:** In Romania, the productivity of the new clones of hybrid poplar has not been tested in recent years. This case study aims to fill a gap on the productivity map of the new poplar clones, estimating, by biomass measurements, the productivity of two clones (AF2 and AF8) with different planting densities (from 1333 trees·ha$^{-1}$ to 2667 trees·ha$^{-1}$). The short-rotation woody crops (SRWCs) were established in homogeneous conditions, in North-East (NE) Romania and the northern part of Suceava County. Using a specifically developed method, biomass production was estimated using destructive methods, with 190 poplar trees being harvested, measured, and weighed to compute the accumulated biomass for each growing season The biomass production of the crops with 1667 trees·ha$^{-1}$ planting density highlighted significant differences in productivity in favour of the AF2 clone after five growing seasons. The crops shared similar annual growth patterns, and the stem biomass represents approximately 73–80% of the total biomass of the trees. The second research question concerning planting density influence on productivity showed fluctuations of biomass accumulations at different planting densities (1333 trees ha$^{-1}$, 1667 trees ha$^{-1}$ and 2667 trees ha$^{-1}$) for a 5-year rotation. The outcomes emphasized the influence of the annual weather conditions— primarily the rainfall in May–June—on poplar growth, showing that productivity also depends on the genotype, density and biotic disturbances.

**Keywords:** biomass production; short rotation forestry; optimal planting density; rotation cycle; SRWCs; hybrid poplar

## 1. Introduction

The current environmental policies support the cultivation of hybrid poplar to produce biomass as a renewable energy source in short-rotation woody crops (SRWCs) [1,2]. The oil shocks of the 1970s triggered the increased interest in these crops [3,4], but the interest is still maintained, considering the Renewable Energy Directive II (part of the European Green Deal) [5] mandates on European countries to achieve a minimum of 32% share of final energy consumption from renewable resources by 2030 [6]. Furthermore, considering the reported increase in $CO_2$ levels in the atmosphere, one effective measure is to augment forest biomass, which can be accomplished by planting currently unforested areas or boosting existing forests' productivity [7,8]. Forestry plays a crucial role in mitigating climate change and securing energy bioresources. As renewable resources, SRWCs positively impact carbon footprint and greenhouse gas mitigation, providing at the same time solid ecosystem services [9].

In recent years, hybrid poplar SRWCs have been identified as a valuable land-use option, considering the biomass yield generates substantial revenue, and the wood might ignite the engine of a sustainable circular bioeconomy. Biomass production in intensive

wood crops is up to 20 Mg·year$^{-1}$·ha$^{-1}$ (ODT, oven-dry tons), or nearly five times more than in some natural forests [10]. Furthermore, biomass yields and crop stability are significantly higher when SRWCs are installed in proper site conditions [11]. Additionally, the resulted crops have the potential to stabilise the biomass market [9].

Aside from genotype and the planting material type, several factors influence biomass production of hybrid poplar crops, such as planting density and row spacing [12–15], length of rotation cycle [16], and also the site and growing conditions (e.g., soil quality, temperature, precipitation, fertiliser administration, or plantation management) [17,18].

Most of these parameters can be controlled by different crop techniques, depending on the nature of the finite product (especially roundwood and woodchips), harvesting method (whole tree or parts of the tree), and annual productivity (using irrigation systems, improving soil properties through amendments and fertilisers, etc.) [19–21].

The planting density impact on short-rotation crop biomass is essential because the distance between trees influences the growth and wood properties [22,23]. The planting density can also affect the canopy closure time and the age of the maximum average growth [24]. Low-density planting schemes favour the increase of branch biomass [25,26]. Moreover, hybrid poplar crops with lower planting density have longer rotation cycles, more flexible harvest periods, and lower installation and maintenance costs [11].

On the other hand, high-density planting schemes can close the canopy quicker [27,28], preventing the development of undesirable competing vegetation and reducing evapotranspiration [29]. However, weed control is necessary for the first year of the crop establishment and after each harvesting cycle [9,30].

The length of the production cycle is a sum of the crop rotations, with 2–4-year rotations being the most frequently used. Lifecycles of 15–20 years are common, but economic analysis of SRWCs shows that longer cycles of 20–25 years can also be feasible [31]. However, the yield diminishes gradually after each rotation [32], which requires additional soil improvement for longer lifecycles. Biomass production is directly proportional to the length of the growing cycle [16,28]; although, the number of individuals per area unit, given the survival rate, decreases in time with the number of harvesting cycles due to the stub's sprouting potential [33].

In addition, the site quality impacts the productivity of short-rotation poplar crops. Reduced biomass accumulation of hybrid poplar is caused by the lack of water and soil nutrients, two main factors influencing tree growth. Therefore, periodic supplying of nutritional supplements [34] and water are critical in the first years of vegetation [35–37]. Poplars are considered heavily water-consuming species [36], generally needing soil moisture that varies between 55% and 60% [13].

According to the FAO (Food and Agriculture Organization), 70 countries encourage the growth of poplar and willow as specialised biomass crops or in combination with other forest species [38]. International Poplar Commission (IPC) reports estimated that the global area cultivated with poplar exceeded 80 million hectares [39]. Short-rotation crop technology is used to manage about 8% of the total cultivated area [40], showing an increased interest in energy crops.

In Europe, the share of renewable energy has increased in recent decades [41]. As a result, woody biomass plays a crucial part in energy generation from renewable sources, accounting for about 70% of total European energy sources [42]. Political support measures and national incentives were primarily used at the European level to support the expansion of SRWCs, an area currently estimated at 50,000 ha [32,43]. Clones of fast-growing species such as poplar or willow are frequently used to establish such crops.

Although studies do not confirm significant advantages of using one of the two genera (*Populus* spp. or *Salix* spp.) for bioenergy plantations in terms of the financial feasibility/production costs [44] or caloric efficiency of their fuelwood [45], in Romania, hybrid poplars seem to be preferred. The hybrid poplar cover grew in Romania, reaching more than 73,000 ha, with approximately 2600 hectares of biomass short-rotation crops [40]. By comparison, hybrid willow crops cover only 800 hectares [46].

Romania has great potential in Eastern Europe for developing SRWCs. An earlier study [47] concluded that Romania has abundant land resources suited for bioenergy production, using growth models to assess the mean productivity of poplar crops to $12.2 \pm 0.5$ Mg·year$^{-1}$·ha$^{-1}$.

Over 800 ha of these new poplar crops are located in the North East (NE) of Romania, in the Suceava county (Rădăuți area) [48,49]. Although there were some negative experiences in the region with monocultures, especially with coniferous and their vulnerability to biotic disturbances [50,51], this reluctance has been reconsidered in recent years. Hybrid poplars and fast-growing species generally present a high bioenergy potential [52], which is now more intensively used. For example, in the study area, poplar biomass is used in a power plant that supplies thermal energy for the city of Suceava.

The source of the planting material for the crops located in the Rădăuți area consisted of rods and cuttings. The goal was to obtain an average annual production of above-ground dry biomass of at least 10 Mg·ha$^{-1}$ (50 Mg per hectare after five growing seasons) [53]. After technical-economic analyses, the landowner set this objective to ensure a minimum level of investment efficiency [53–55]. However, further investigations were conducted to identify the main factors leading to lower biomass production, focusing on planting spacing, which is the easiest feature to adapt in the crop management system [56]. A few decades ago, Romania had solid experience in cultivating poplars, but the productivity of the new clones has not yet been tested in recent years. This case study aims to fill a gap on the productivity map of the new poplar clones in Europe, estimating, by biomass measurements, the productivity of two clones (AF2 and AF8) with different crop planting densities.

The main purpose of this research is to evaluate the factors that significantly contribute to increasing biomass production in short-rotation poplar crops in the NE of Romania, an area with climatic particularities and conditions that can lead to significant differences in biomass productivity compared to other regions. Furthermore, to better understand the productivity mechanisms, our goal was to provide answers to the following research questions:

1.  What is the biomass production of the two analysed clones for the hybrid poplar crops with the most common planting density (1667 trees·ha$^{-1}$) after five growing seasons?
2.  What are the differences in biomass accumulations of hybrid poplar crops at different planting densities?

## 2. Materials and Methods

### 2.1. Site Location

The study was conducted in hybrid poplar crops installed with two clones, AF2 (*P. deltoides 145-86 × P. nigra 40*) and AF8 (*P. × generosa 103-86 × P. trichocarpa PEE*), at different densities and ages (Table 1). The crops were located in the NE of Romania, in the northern part of Suceava County (Dornești, Vicșani and Fântâna Mare), close to the Ukrainian border. The area has a dominant temperate continental climate, characteristic of the Suceava Plateau [57,58]. The average annual temperature is between 7 °C and 8 °C, with an average yearly rainfall of approximately 550–600 mm [59]. The variations in temperature and precipitations were recorded for the analysed time frame (Figure S1). The soil type is *Faeoziom cambic*, with high productivity and uniform texture [60,61].

### 2.2. Data Collection

To achieve research objectives, ten short-rotation hybrid poplar plantations/crops located in NE Romania were studied. The crops were homogeneous in terms of climatic conditions, soil type and installation method, but there were differences in size, density, and age (Table 1). The size of the crop plots varies between 0.11 and 13.8 hectares.

The same installation method was used for all the plots. Poplar trees were propagated vegetatively using rods of 180 cm with 2–3 cm in diameter. The rods were inserted 0.6 m deep. The planting distance was 3 m between rows for all the poplar crops to allow mechanised maintenance. The distance between individuals on the same row was 1.5 (2222 trees·ha$^{-1}$), 2 m (1667 trees·ha$^{-1}$), or 2.5 m (1333 trees·ha$^{-1}$).

**Table 1.** Summary data of the analysed poplar crops.

| No. | Clone | Year of Establishment | Location | Age (Years) | Size (Ha) | Density (Trees·ha$^{-1}$) | Number of Trees | Coordinates | |
|-----|-------|-----------------------|----------|-------------|-----------|---------------------------|-----------------|-------------|---|
| 1 | AF8 | 2013 | Fântâna Mare | 3 | 4.22 | 1667 | 10 | 47°53′47.34″ N | 26° 0′46.05″ E |
| 2 | AF8 | 2013 | Fântâna Mare | (2) 3 | 0.33 | 2222 | (10) 10 | 47°53′38.86″ N | 26° 0′30.48″ E |
| 3 | AF2 | 2013 | Fântâna Mare | (2) 3 | 0.33 | 2222 | (10) 10 | 47°53′38.86″ N | 26° 0′30.48″ E |
| 4 | AF8 | 2011 | Fântâna Mare | 5 | 13.81 | 1667 | 30 | 47°53′56.15″ N | 26° 0′35.11″ E |
| 5 | AF2 | 2011 | Fântâna Mare | 5 | 4.14 | 1667 | 30 | 47°53′41.40″ N | 26° 0′42.67″ E |
| 6 | AF2 | 2011 | Vicșani | 5 | 7.14 | 1667 | 10 | 47°55′35.27″ N | 25°59′10.48″ E |
| 7 | AF8 | 2010 | Dornești | 6 | 0.58 | 1667 | 10 | 47°51′11.63″ N | 25°58′53.49″ E |
| 8 | AF2 | 2009 | Dornești | (4) 5 | 0.11 | 2667 | (20) 20 | 47°50′53.78″ N | 25°57′59.09″ E |
| 9 | AF2 | 2009 | Dornești | 5 | 0.11 | 1333 | 10 | 47°50′51.89″ N | 25°58′0.44″ E |
| 10 | AF8 | 2009 | Dornești | 5 | 0.11 | 1333 | 10 | 47°50′51.32″ N | 25°58′2.78″ E |

Site preparation started the year before planting and included weeding and mechanical tillage, and the soil was also harrowed the following spring. Maintenance was done in the first two growing seasons with two tillages between rows per year, and herbicides were applied to control weeds.

Crop biomass production was estimated using destructive methods, with 190 poplar trees being harvested, measured, and weighed [62]. At least ten trees were extracted from each crop. The individuals were systematically removed (every 5th tree) from the middle rows to avoid the anomalies induced by the edge effect. More than ten trees were extracted in larger crops, where the distance from the edge allowed it. The trees were harvested outside of the growing season when the leaves had already fallen.

The biomass was determined based on the wood's mass, volume, and humidity for each individual. The annual biomass was computed based on the stem volume for each growing season, using a methodology developed in earlier studies which reconstructs the annual volume of the stem using multiple diameters and height records (Figure 1) [48,49].

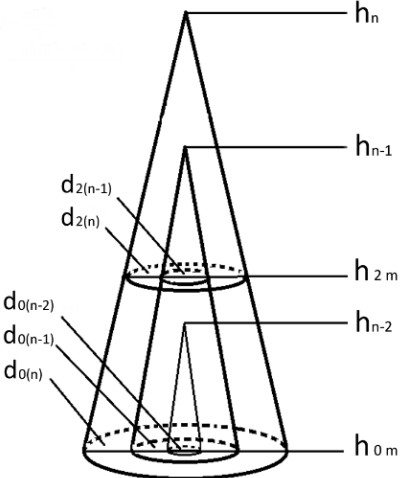

**Figure 1.** The biometric features used for stem volume estimation for each growing season.

The field activity included felling and weighing the trees, cleaning branches and re-weighing the cleansed trunk, measuring the total length of the stem and the annual height growths, sectioning the stem to 2 m, measuring the lengths and diameters of

the parts, removing three discs (from the base, 1 and 2 m) used for annual growth and humidity assessment.

Subsequently, the discs were prepared in the laboratory for humidity assessment and then sanded. After scanning, the tree rings measurements were performed to determine the annual diameters of the basal and superior piece of the stem.

The stem volume was calculated without bark by assimilating it with a cone. The stem, being physically divided into two separate parts, allowed accurate measurements of biometric dimensions. The first part, consisting of the first 2 m of the basal section, was geometrically assimilated to a cone trunk, and the second part, consisting of the rest of the stem, was assimilated to a cone. The following equation was used (Equation (1)).

$$V_{1,2} = \frac{\pi}{12}\left(d_1^2 + d_2^2 + d_1 d_2\right)h \tag{1}$$

where $V_{1,2}$ (m$^3$) is the volume of the stem parts (for which $V_1 + V_2 = V_t$); $V_1$ is the basal stem's volume with a length of 2 m; $V_2$ is the volume of the superior part of the stem up to the terminal bud; $d_1$ is the diameter of the first section of the stem; $d_2$ is the diameter of the second section of the stem; and h is the total length of the stem.

The above-ground biomass was estimated for the stem and branches, taking into account the relative moisture of the harvested samples for each tree part (Equation (2)), and represented annually in relation to stem volume dynamics [63,64].

$$B = \frac{Mt_v}{1+U} \tag{2}$$

where $B$ is the dry weight of the biomass of the tree part (kg); $Mt_v$ is the total weight of the tree part in its fresh state; and $U$ is the specific moisture of the tree part (%).

Tree total and annual heights were identified and measured on the tree stem after harvesting using a measuring tape with an accuracy of 0.01 m. Diameters (cm) were measured on the scanned images of discs using CooRecorder and C.Dendro 7.8 software [65] in two perpendicular directions per wooden disc, with an accuracy of 0.01 cm. The discs were harvested as moisture samples at different stem heights (at the base, at 1 and 2 m).

The collected samples were dried in the oven at 105 °C to a constant weight [23,56]. Thus, the relative moisture of each sample (%) was the result of the two-weighing difference in fresh and dry states (Equation (3)) [66]. The stem's moisture was determined as an average of the stem's sample wood discs moisture, collected from the base, and at 1 and 2 m height. For the branches, the humidity was determined using a randomly selected medium-sized branch.

$$U = \frac{M_v - M_u}{M_u} \times 100 \tag{3}$$

where $U$ is the specific moisture of the tree parts; $Mv$ is the weight of samples in their fresh state; and $Mu$ is the weight of samples in dry condition. The sample's average biomass was related to the surface unit (in Mg·ha$^{-1}$) according to the crop density [67]. However, the results, expressed for each corresponding crop density, can be adjusted with specific survival rates of the crops.

### 2.3. Data Analysis

The data analysis was carried out using XLStat 2012. The influence of the genotype on biomass accumulation was tested using the non-parametric Kruskal–Wallis test. The annual biomass was analyzed with respect to the clone from a specific location. Then, biomass per hectare of the tree parts (after five growing seasons) was analyzed with respect to the clone from a particular site. We decided to use a non-parametric test, even though the tested distributions were normal according to the Shapiro–Wilk test. However, the condition of homogeneity of variance was not met, according to Levene's test. Bonferroni–Dunn's multiple comparisons post hoc analysis [68] was used to assess the significance of the means differences.

## 3. Results

### 3.1. The Biomass Production of Hybrid Poplar Crops after Five Growing Seasons

Most of the poplar SRWCs installed in the NE of Romania use a planting spacing of $2 \times 3$ m (1667 trees·ha$^{-1}$). Therefore, we selected the crops with this planting density and a minimum rotation of 5 years to answer the first research question. Three hybrid poplar crops were analysed (lines 4 to 6, Table 1), having the same density of 1667 trees·ha$^{-1}$ (spaced at $3 \times 2$ m), and being installed in the same year (2011). In the Vicșani area, the clone AF2 was used, and in Fântâna Mare, the clones AF2 and AF8.

This particular planting scheme, with a 1667 trees·ha$^{-1}$ density, led, after five growing seasons, to large quantities of biomass, ranging from 31.76 to 46.01 Mg·ha$^{-1}$ in the specific climatic conditions of the Dornești-Fântâna Mare-Vicșani area for the 2011–2015 period.

In the first three years (2011–2013), there were no significant differences in biomass accumulation per hectare between the crops. However, beginning in 2014, the biomass accumulation of the Vicșani crop was significantly superior to the other two crops from Fântâna Mare. Statistically, significant differences were observed in 2014 with respect to the AF8 clone from Fântâna Mare, and also in 2015 with respect to both clones from the same area (Table 2).

**Table 2.** The total biomass per hectare (Mg) cumulated after five growing seasons (average ± s.d.).

| Year | AF2–Vicșani | AF2–Fântâna Mare | AF8–Fântâna Mare | Kruskal–Wallis | |
|------|-------------|------------------|------------------|-----|-----|
| | | | | *K* | *p* |
| 2011 | 0.74 ± 0.3 [a] | 0.74 ± 0.22 [a] | 0.64 ± 0.37 [a] | 1.8981 | 0.3871 |
| 2012 | 5.23 ± 1.4 [a] | 4.53 ± 1.69 [a] | 3.78 ± 1.63 [a] | 2.2282 | 0.282 |
| 2013 | 19.42 ± 3.16 [a] | 16.66 ± 4.02 [a] | 14.19 ± 5.26 [a] | 3.6262 | 0.1631 |
| 2014 | 34.53 ± 3.82 [a] | 30.17 ± 6.16 [a] | 26.19 ± 4.82 [b.] | 5.9915 | 0.0002 |
| 2015 | 45.86 ± 3.88 [a] | 38.27 ± 8.05 [b] | 31.53 ± 5.96 [c] | 5.9915 | <0.0001 |

**Note**. s.d. is the standard deviation; a, b, c, are the significance levels of the statistical test. Values with the same letter do not differ significantly (according to the Bonferroni–Dunn's post hoc analysis for *p* < 0.05); *K* is the statistic from the Kruskal–Wallis; *p* is the statistical significance of the results.

Regardless of the area studied, the annual mean increase in biomass was 7.71 Mg·ha$^{-1}$. The AF2 clone from the Vicșani crop recorded the highest annual average increase in biomass, in 2014, with approx. 15.11 Mg·ha$^{-1}$ (32.9% of the total biomass production).

The annual biomass growth for the three analysed crops was ascendant until 2014, but significantly reduced in 2015 (Figure 2). The lowest values are recorded by the AF8 clone in the Fântâna Mare area. The highest growth contribution for this clone and density was recorded also in 2014, with approx. 38% (12 Mg·ha$^{-1}$). The fifth year's contribution to total production ranged from 17% (for clone AF8 in Fântâna Mare) to 24.7% (AF2–Vicșani).

The above-ground biomass distribution on the main parts of the tree (stem and branches) was also assessed (Table 3). The branches represent approximately 20% (AF2 clone from Vicșani and AF8 clone from Fântâna Mare) and 27% (AF2 clone from Fântâna Mare) of the total analysed tree biomass.

**Table 3.** The biomass per hectare accumulated on tree parts after five growing seasons (Mg·ha$^{-1}$, mean ± s.d.).

| Tree Parts | AF2–Vicșani | AF2–Fântâna Mare | AF8–Fântâna Mare | Kruskal–Wallis | |
|------------|-------------|------------------|------------------|-----|-----|
| | | | | *K* | *p* |
| stem | 37.01 ± 3.46 [a] | 28.22 ± 5.59 [b] | 25.28 ± 5.29 [b] | 24.7105 | <0.0001 |
| branches | 8.99 ± 1.05 [a] | 10.28 ± 3.38 [a] | 6.48 ± 1.43 [b] | 27.7793 | <0.0001 |
| total | 45.86 ± 3.88 [a] | 38.27 ± 8.05 [b] | 31.53 ± 5.96 [c] | 24.5634 | <0.0001 |

**Note**. s.d. is the standard deviation; a, b, c, are the significance levels of the statistical test. Values with the same letter do not differ significantly (according to Bonferroni–Dunn's post hoc analysis, for *p* < 0.05); *K* is the statistic from the Kruskal–Wallis; *p* is the statistical significance of the results.

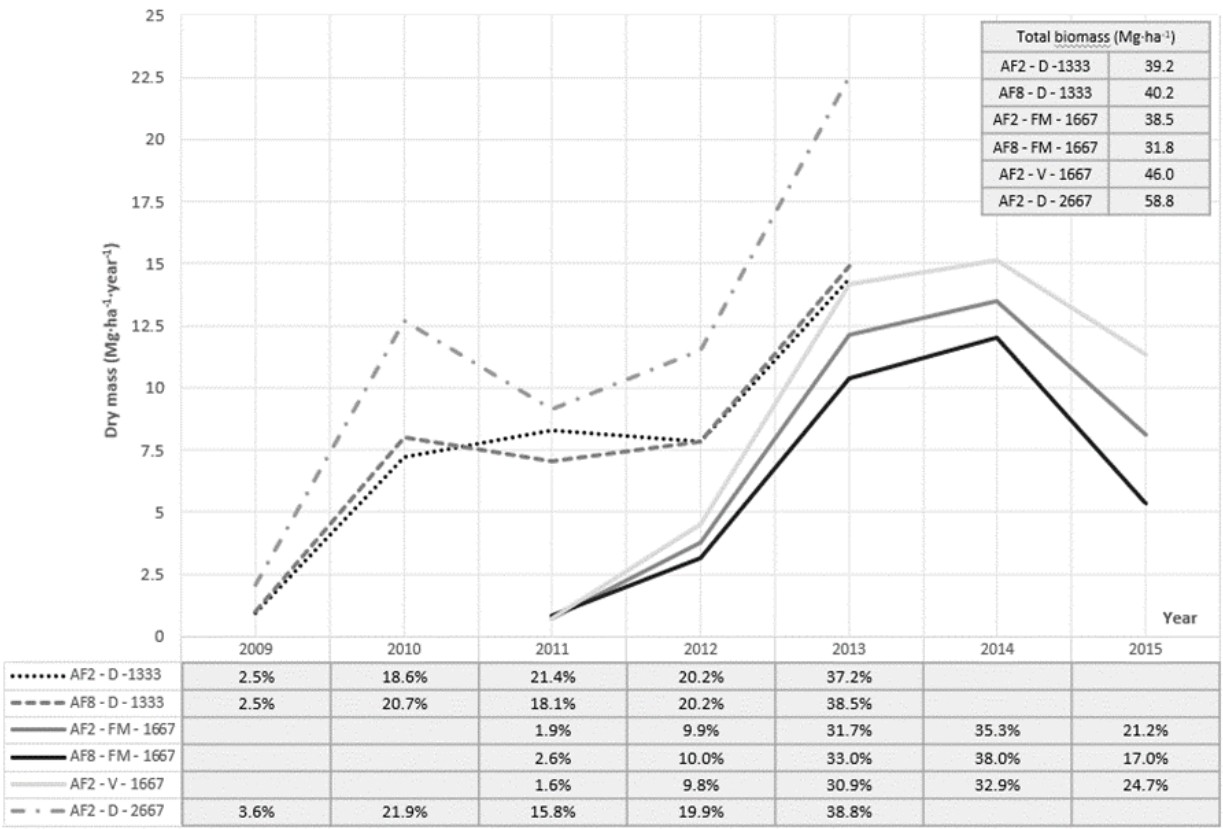

| | Total biomass (Mg·ha⁻¹) |
|---|---|
| AF2 - D -1333 | 39.2 |
| AF8 - D - 1333 | 40.2 |
| AF2 - FM - 1667 | 38.5 |
| AF8 - FM - 1667 | 31.8 |
| AF2 - V - 1667 | 46.0 |
| AF2 - D - 2667 | 58.8 |

| | 2009 | 2010 | 2011 | 2012 | 2013 | 2014 | 2015 |
|---|---|---|---|---|---|---|---|
| AF2 - D -1333 | 2.5% | 18.6% | 21.4% | 20.2% | 37.2% | | |
| AF8 - D - 1333 | 2.5% | 20.7% | 18.1% | 20.2% | 38.5% | | |
| AF2 - FM - 1667 | | | 1.9% | 9.9% | 31.7% | 35.3% | 21.2% |
| AF8 - FM - 1667 | | | 2.6% | 10.0% | 33.0% | 38.0% | 17.0% |
| AF2 - V - 1667 | | | 1.6% | 9.8% | 30.9% | 32.9% | 24.7% |
| AF2 - D - 2667 | 3.6% | 21.9% | 15.8% | 19.9% | 38.8% | | |

**Figure 2.** The annual accumulation of dry biomass and its share of the total yield considering the different densities, clones, and crop location (FM–Fântâna Mare; V–Vicşani; D–Dorneşti). Note: The values from the table below the graph show the cumulative biomass percentage of each growing season from the total, and the table on the top right shows the total biomass at the end of the five growing seasons.

Significant differences in this distribution are also observed between the crops. The Vicşani AF2 clone differs from the other two clones installed at Fântâna Mare, considering the stem biomass share. At the end of the five growing seasons, the highest stem biomass production was 37.01 Mg (80.4%) (AF2–Vicşani), and the lowest was recorded by the Fântâna Mare AF8 clone with 25.28 Mg (79.6%). However, this pattern is not consistent. In the case of branches, significant differences are recorded across clones, regardless of their location. The AF8 clone has the lowest production for this component (6.48 Mg·ha⁻¹), with a share of only 20.4% of the total biomass, while the AF2 clone from Fântâna Mare has the biggest share (26.7%).

### 3.2. The Biomass Accumulations of Hybrid Poplar Crops at Different Planting Densities

Finding the optimal planting density for poplar SRWCs is a matter of debate, considering the more significant costs associated with higher densities and the mechanisation difficulties. The second research objective focused on three specific planting densities: 1333 trees·ha⁻¹, 1667 trees·ha⁻¹ and 2667 trees·ha⁻¹, with the same planting distance between rows (3 m) to allow mechanised crop maintenance. We selected the crops with the same 5-year rotation to compare their productivity.

The trend of annual biomass accumulations is generally increasing. However, there are significant differences between the crop groups with different densities, as shown in Figure 2. The crops with a density of 1667 trees·ha⁻¹ (planted in 2011) maintain an ascending trend only for the first four years (2011–2014), while the crops with other densities (planted in 2009) prolong their accumulation ascending tendency until the fifth year.

The various growth patterns can be explained by the difference in planting density and weather conditions. The crops were installed in different years (2009 for the crops with 1333 and 2667 trees·ha$^{-1}$, respectively; 2011 for the crops with 1667 trees·ha$^{-1}$), facing specific weather conditions in different growing seasons. In the spirit of this idea, the graph shows that the crops installed in 2009 have a slight decline or stagnation in accumulations in 2011 and 2012, compared with 2010. However, we could interpret this fact from a different perspective by analyzing the weather data from those years (Figure S1). Taking into account the rainfall in May and June, the annual periods with the maximum growth of poplar [69], we noticed that 2010 was an exceptional year, with cumulated rainfall in the two months of over 350 mm. These higher amounts led to higher biomass accumulations in 2010, compared with the following two seasons. A common steep trend found in all the analyzed crops, which similar weather conditions may have influenced, is the increase in biomass accumulations in 2013, another year with significant precipitation in May and June.

All the crops planted in 2009 (with a density of 1333 trees·ha$^{-1}$ and 2667 trees·ha$^{-1}$) had the maximum growth in 2013, with more than 37% of the total yield accumulated during the fifth season. Surprisingly, the crops with a density of 1667 trees·ha$^{-1}$, planted in 2011, reached the maximum growth earlier, in the fourth season (2014), with 32.9%–38.0% of the total biomass. Moreover, the growths from the fifth season of these latter crops (17.0%–24.7%) were even smaller than the growths from the third season (30.9%–33.0%).

The differences in productivity between crop densities are best emphasised by analysing the data for the crops installed in the same year. The crops with densities of 1333 and 2667 trees·ha$^{-1}$ were established in 2009; thus, we have homogeneity, with approximately the same site conditions and weather conditions specific to each growing season (Figure S1). The results plotted in Figure 2 show superior annual values and total productivity (58.82 Mg·ha$^{-1}$) of the AF2 Dornești crop with a density of 2667 trees·ha$^{-1}$, compared to AF2 and AF8 Dornești crops with a density of 1333 trees·ha$^{-1}$ (39.16 and 40.18 Mg ha$^{-1}$, respectively).

Considering the 5-year rotation, significant differences were noted in the productivity of the crops with different densities. The AF2 Dornești crop, with a density of 2667 trees·ha$^{-1}$, recorded the largest production per year and hectare (11.4 Mg·ha$^{-1}$·year$^{-1}$). On the other hand, AF2 Vicșani, with a density of 1667 trees·ha$^{-1}$, delivered the maximum yield for the crops (9.2 Mg·ha$^{-1}$·year$^{-1}$). The crops with a density of 1333 trees·ha$^{-1}$, AF2 and AF8 Dornești, provided close but lower production values of 7.8 and 8.0 Mg·ha$^{-1}$·year$^1$, respectively.

The increasing trends of biomass accumulation during the rotation period are noticeable for the selected crops. Therefore, the dynamics of crop annual productivity at a specific planting density can provide valuable insights into the decision-making process, such as choosing the rotation length to maximize biomass production. Poplar SRWCs from NE Romania generally have 3- or 5-year rotation periods, with most of the studied crops falling within this range. Based on our study measurements and results, and using bibliographic references regarding the productivity of poplar SRWCs with similar rotation lengths (from Supplementary Materials, Table S1), we have plotted a chart (Figure 3) showing the average annual growth per tree at different planting densities.

We estimated the average annual growth per individual for particular crop densities based on the regression equation (Table 4). Then, we assessed the biomass production per year and hectare, considering the number of trees per hectare (tree density). Finally, we projected the total biomass production for the entire 5-year rotation period. The obtained results show a biomass accumulation of over 50 Mg·ha$^{-1}$ in the range of densities of 833–1667 trees·ha$^{-1}$. Individual biomass increases from a high-density planting scheme to a low-density planting scheme, but this growth is corrected by reporting individual quantities per area [25,70].

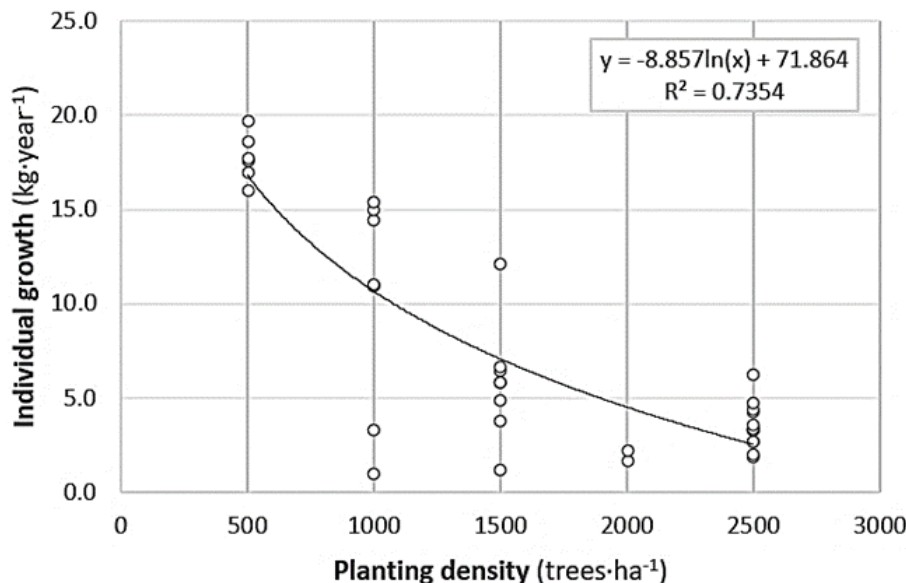

**Figure 3.** The average annual growth per individual at different planting densities. Note: The planting density ranges from 500 to 2500 trees·ha$^{-1}$, rotation length from 3 to 6 growing seasons, and rainfall from 400 to 1200 mm.

**Table 4.** The estimation of the biomass yield per hectare (Mg·ha$^{-1}$) for different planting densities.

| Spacing (m × m) | Density (Trees·ha$^{-1}$) | Yield (Mg·ha$^{-1}$·Year$^{-1}$) | Total Biomass in 5 Years (Mg·ha$^{-1}$) |
|---|---|---|---|
| 3 × 1.25 | 2667 | 5.32 | 26.59 |
| 3 × 1.5 | 2222 | 8.02 | 40.11 |
| 3 × 2 | 1667 | 10.26 | 51.31 |
| 3 × 2.5 | 1333 | 10.85 | 54.23 |
| 3 × 4 | 833 | 10.25 | 51.24 |
| 4 × 4 | 625 | 9.28 | 46.39 |

## 4. Discussion

### 4.1. Biomass Production Assessment after Five Growing Seasons for the Crops with the Most Common Planting Density

The current investigation results indicate statistically significant differences in the total biomass yield of the analysed crops, recorded for the 1667 trees·ha$^{-1}$ planting density, at the end of the 5-year rotation, under the climatic conditions of NE Romania (Table 2). Still, certain similarities in the dynamics of annual biomass accumulations of crops were also observed (Figure 2).

The determinant variables of the differences in the total biomass production can be considered the local site conditions, the clones' different genotypes, and the possible disturbing factors that could have occurred during that 5-year period, such as defoliator attacks or extreme weather conditions.

Regarding the different behaviour of the two Italian cultivars (AF2 and AF8), no significant growth differences were noticed in the first three years, with the total biomass per hectare being similar between 2011 and 2013 for all the crops. However, differences emerged in the fourth year (2014), when both crops with AF2 clones (AF2–Vicşani and AF2–Fântâna Mare) detached from the AF8 crop's production (AF8–Fântâna Mare). In the 5th year, this trend was preserved and even accentuated, with significant differences in production for all the crops. However, both crops with AF2 clones maintained their lead (with 45.86 Mg·ha$^{-1}$ for Vicşani and 38.27 Mg·ha$^{-1}$ for Fântâna Mare) over AF8 productivity (31.53 Mg·ha$^{-1}$). The poorer productivity performance of the AF8 cultivar is

also highlighted in the last two seasons compared to AF2, located at the same site (Fântâna Mare), thus sharing similar conditions.

AF2 and AF8 are two new cultivars used in Romania's commercial energy crops. Unfortunately, no other studies have been conducted in other areas of Romania to compare the productivity data of these Alasia Franco cultivars (AF). However, the results of our research concerning these two clones' productivity are strongly correlated with other studies and publications from abroad. Although both clones are known to be very productive, with minor differences in productivity under Italian climatic conditions [28,71], it appears that in other conditions, these gaps between the two clones are widening. In many comparative studies involving both clones, in different geographical conditions at a higher latitude, e.g., Poland [72,73] or Latvia [74], the cultivar AF2 was a more productive genotype than AF8. Furthermore, Niemczyk [73] suggests these new cultivars might exhibit a phenological mismatch to the northern climatic conditions, being more affected by frost damage and insect attacks than the other cultivars.

The differences in productivity recorded in this study can be partially explained by different crop locations involving slightly different sites or weather conditions. The crops located in the Fântâna Mare area performed less, which could be explained by some defoliator' attacks (*Clostera anastomosis*). The moth developed a strong presence in AF2 and AF8 crops from Vicșani and Fântâna Mare, starting in 2013 [75].

Regarding the influence of the different locations of the crops, they are positioned close to each other, sharing approximately the same site and weather conditions. However, the factors that showed a strong presence in the crops at Fântâna Mare (defoliator's attacks, an intense hail storm in June 2012, or the slightly different soil texture conditions), compared to Vicșani, could have led to a drop in biomass for both clones.

The productivity of fast-growing species is, in general, directly influenced by soil water supply [36]. Therefore, minor variations of this factor could cause substantial productivity differences because adequate water resources are essential, especially in the early years of crop growth for poplar [18].

Nevertheless, the productivity results obtained in this study after three growing seasons are comparable not only to other poplar studies, but also to other fast-growing species (e.g., *Eucalyptus* spp.), where biomass varied between 15–22 Mg·ha$^{-1}$ for the most favourable conditions [76].

The distribution of annual biomass accumulations during the five-year rotation is also important because it might reveal particular growth patterns under certain conditions. All the crops shared approximately the same pattern, with minor differences highlighted in Figure 2. The growth trend was upward during the first three years for all crops, but then the accumulation tempered its rise in the fourth season, and even decreased in the fifth season.

The biomass added in the fifth year ranged from 16.9% (AF8–Fântâna Mare) to 24.7% (AF2–Vicşani). These values are significantly lower than those obtained for the same cultivars in similar conditions during 2009–2013, when the biomass accumulation in the fifth year was approx. 36% [48]. Different annual weather conditions most probably cause these differences. An extreme drought occurred at the beginning of the growing season in 2015, during May and June, the annual periods with the maximum growth of poplar. For these two months, the precipitations recorded at the Rădăuți weather station were 24.4 L·m$^{-2}$ and 44.8 L·m$^{-2}$, respectively (Figure S1). In 2013, the fifth growing season of the crop with similar conditions, the recorded rainfall was 102.4 L·m$^{-2}$ in May and 130.8 L·m$^{-2}$ in June, quantities which could have boosted biomass accumulation in that particular year.

### 4.2. The Biomass Production of Hybrid Poplar Crops at Different Planting Densities

The planting density directly influences the biomass production, but also the possibility of mechanisation of installation and maintenance activities, significantly impacting the cost of these activities. Most of the hybrid poplar crops from the NE of Romania use a density

of 1667 trees·ha$^{-1}$, with trees spaced at 3 × 2 m. We analysed AF2 and AF8 crops with different densities (1333 trees·ha$^{-1}$, 1667 trees·ha$^{-1}$, and 2667 trees·ha$^{-1}$), but the same planting distance between rows (3 m) and rotation length (five years), to compare their productivity. The typical planting design with 3 m between the rows allows for mechanised maintenance [54]. If a larger distance is chosen between individuals, paying more attention in the early growing seasons is necessary to remove the competing vegetation, which involves additional costs [28,77].

Our results showed different biomass accumulation patterns depending on the genotype, density, and the variable annual weather conditions. Moreover, the rotation length directly influences the SRWCs biomass yield. Therefore, different rotation periods are often tested under various conditions to obtain the best outcomes. The dynamics of the crops' annual productivity at a specific density can be used to choose a certain rotation period to provide the highest biomass production. That's because, in terms of establishing the rotation, the management strategy aims for a harvest age at which the average growth is maximum, or when its trajectory starts to decline [78].

Generally, according to the literature, SRWC should be harvested in the following order: crops with planting densities between 10,000 and 14,000 trees·ha$^{-1}$ should be harvested after one season of vegetation (also known as coppice crops). The crops with planting densities between 5000 and 6000 trees·ha$^{-1}$ should be harvested after two or three years, and the ones with planting densities between 1000 and 1800 trees·ha$^{-1}$ should be harvested after five growing seasons [71,74].

For example, in our analysis, the crops planted in 2009 had the maximum growth in the fifth year, with more than 37% of the total yield accumulated during this last season. This pattern of annual biomass accumulations shows a solid growth potential, which could lead, from a managerial point of view, to a decision to extend the rotation period to maximise production. In addition, some studies investigated this topic and displayed the benefits of this increase [9,31].

Furthermore, in the case of the crops selected for this study, we also analysed a 6-year crop (AF8 Dornești—Table S2, line 7), where, during the fifth year, the proportion of accumulated biomass was approx. 37.3%. Because of severely reduced precipitation in the maximum vegetative accumulation period (May–June) of the sixth year, the crop accumulated biomass of approx. 22.3 Mg·ha$^{-1}$, which represents approx. 34.2% of the total production. Still, this crop reached a yield of approx. 10.8 Mg·ha$^{-1}$·year$^{-1}$, with 64.9 Mg·ha$^{-1}$·year$^{-1}$ after six growing seasons, which might weigh heavily in deciding about prolonging rotation. It has been shown that increasing the rotation above 7–8 years is not necessarily economically advantageous [79]. Typically, these SRWC are harvested at 3–5 years, and have a 15 to 30 year life cycle, depending on the site conditions [80].

On the other hand, the crops with a density of 1667 trees·ha$^{-1}$, planted in 2011, reached the maximum growth in the fourth season. Surprisingly, the accumulations in the fifth season (2015) of these crops were even smaller than those in the third year (2013).

Of course, 2013 was very rich in May–June precipitation, and allowed significant biomass increases for all clones. Still, outcomes may be influenced by the different densities and conditions of 2015, when defoliator attacks were reported [81].

The production projection based on the regression equation at different densities (Table S1) shows an optimum of the total production at densities between 833 and 1667 trees·ha$^{-1}$ (with a maximum of 1333 trees·ha$^{-1}$). However, our results emphasised the productivity gaps between crops with different densities for the crops planted in 2009, the crops with densities of 1333 and 2667 trees·ha$^{-1}$. The outcomes showed superior yearly biomass accumulations and total productivity of the crop with a density of 2667 trees·ha$^{1}$ (AF2–Dornești) compared to crops with a density of 1333 trees·ha$^{-1}$ (AF2 and AF8–Dornești). This indicates that significant variations in production can be recorded depending on the weather conditions (primarily the rainfall in May–June), the fast-growing species being very sensitive productively to this aspect [17,18,34]. Definitely, other disturbances can affect biomass

accumulations, especially biotic ones. In this case, poplar insect attacks (*Clostera anastomosis*) were recorded in the studied area in that particular period [82].

On the other hand, at a planting scheme smaller than $3 \times 1.25$ m, after five years, for clone AF2, an individual biomass of 22.05 kg was obtained in an earlier study [48,54]. The Fântâna Mare crop produced a similar quantity after the same period for the $3 \times 2$ m scheme, with 24.22 kg·tree$^{-1}$. In this context, the denser planting scheme led to a biomass gain per hectare of approx. 18.4 Mg·ha$^{-1}$. Low-density crops favour a higher percentage of biomass obtained from branches. Thus, the relationship between the branch and the stem biomass changes by increasing the planting distance [15,25,83].

Regarding the clone's productivity, we pointed out that, in the case of crops with a density of 1667 trees·ha$^{-1}$, the differences between the productivity of cultivars are significant, with AF2 being the more productive clone. These differences were not observed for these cultivars in the case of crops from Dorneşti, with a planting density of 1333 trees·ha$^{-1}$, installed in 2009. The productivity of these crops is similar, following approximately the same pattern of the annual distribution of biomass accumulations. Further studies are needed to evaluate particular clone biomass production at different planting densities, for specific sites.

Further, according to some studies [79], the quality of the plant material is critical for biomass production in poplar crops, at least for $2 \times 2$ m, $3 \times 2$ m, and $3 \times 3$ m spacing. Therefore, the crops installed with planting material from cuttings or rods of one year need one (for 1-year rods) or two (for cuttings) additional years to obtain the same yield as the crop planted with rods of two years (Table S3).

Rotation periods of 3 or 4 years, tested for different genotypes under less favourable site conditions, were evaluated in Spain [84]. The results showed that biomass yield increases from 74% to up to 166% when rotation length is extended by one year in most genotypes. In the current investigation, we found similar results for the fourth growing season of the crops planted in 2009. As a result, the rotation should be selected considering both the genotype and site conditions, with longer rotation being advised when conditions are less favourable. Still, further analysis is needed to determine the best rotation length for a particular site, cultivar, and management technique.

## 5. Conclusions

The case study focused on the biomass production of short rotation poplar crops based on two representative selected genotypes frequently used in Europe (AF2 and AF8). The results of biomass production after five growing seasons for the crops with the most common planting density (1667 trees·ha$^{-1}$) highlighted differences in productivity between the two analysed clones in favour of the AF2 clone. These findings underline the importance of genotype selection in establishing a poplar plantation.

Still, this research raises more questions for further investigation related to planting density's impact on crop productivity. Our results indicate a higher production at 2667 trees·ha$^{-1}$ planting density; however, to supplement our findings, we need to analyse additional crops installed in the same year, to ensure similar weather conditions for each growing season.

Since the crops are homogeneous in terms of climatic conditions, soil type, installation method, and treatment, the differences in biomass production between the plantations with different densities could be explained by different weather conditions (primarily the rainfall in May–June) or biotic disturbances. Overall, the present case study provides valuable information and a framework for future studies to assess the productivity of hybrid poplar SRWCs in Romania. Considering the uncertainties and the effects of climate change, landowners need more accurate information about the productivity of SRWCs in specific conditions before investing in them, because they want to keep the production cost per biomass unit obtained as low as possible. In the last decade, there have been several funding programs to support these types of crops, but lately, the availability of specific

public incentives has decreased. As a result, some existing SRWC plantations have been converted to more profitable crops.

However, the recent energy crisis highlights the need to identify sustainable bioenergy sources, bringing SRWCs back into the spotlight. In this context, improving crop technologies is vital to achieving profitable and sustainable SRWC. Europe may be interested in biomass produced using SRWCs, since it provides a solid option for establishing a circular bioeconomy while delivering significant environmental services.

**Supplementary Materials:** The following are available online at: https://www.mdpi.com/article/10.3390/f13071089/s1, Figure S1: Precipitation and annual temperatures recorded at the Suceava Weather Station (located about 30 km south of the poplar crops areas); Table S1. Biomass yield obtained for different rotation length, planting density and growing conditions; Table S2. Summary table of data; and Table S3. Yield of short-rotation crops with different planting densities and rotation. References [85–95] are cited in the Supplementary Materials files.

**Author Contributions:** Conceptualisation, I.-C.D. and C.P.; data curation, I.-C.D.; formal analysis, I.-C.D., C.M. and C.P.; funding acquisition, I.-C.D.; investigation, I.-C.D.; methodology, I.-C.D., C.P. and C.M.; project administration, I.-C.D.; resources, I.-C.D.; software, I.-C.D. and C.P.; supervision, C.P.; validation C.P.; visualization, I.-C.D.; writing—original draft, I.-C.D., C.M. and C.P.; writing—review and editing, I.-C.D. and C.P. All authors have read and agreed to the published version of the manuscript.

**Funding:** This research was funded by a grant of the Romanian Ministry of Education and Research, CNCS-UEFISCDI, project number PN-III-P1-1.1-PD-2019-0388, project ForCrops (PD3/2020).

**Institutional Review Board Statement:** Not applicable.

**Informed Consent Statement:** Not applicable.

**Data Availability Statement:** Not applicable.

**Acknowledgments:** We thank Mihai-Leonard Duduman, Marcel Vlad Hazi, and Emil Gheorghe for their help in collecting field data. We wish to thank the anonymous reviewers for their valuable and constructive recommendations for the improvement of the paper.

**Conflicts of Interest:** The authors declare no conflict of interest. The funders had no role in the design of the study, in the collection, analysis, or interpretation of data, in the writing of the manuscript, or in the decision to publish the results.

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
