# Peer review of "Productivity of Short-Rotation Poplar Crops: A Case Study in the NE of Romania"

_forests, doi:10.3390/f13071089_

Round 1
Reviewer 1 Report
The introduction is tedious and the objectives are not clearly defined.
Line 106 and 108: I don't detect the difference between both questions, I should clarify it more.
Line 173: B is the dry weight of the biomass? should clear it up
Line 197-198: The statistical analysis is not clear. It must indicate which variables were studied and with respect to which factors. It must indicate which variables followed normal distributions and equality of variances and which did not.
There are other factors besides the plantation density that influence the amount of biomass accumulated (Fertilization, Silvicultural treatments,...)
Tables from other studies should not be reproduced in the discussion, the reference is sufficient.
The conclusions must be supported by the results and references must be avoided in them, as well as discussions.
Some references are difficult to access
Author Response
Dear Reviewer,
We thank you for the effort and the time spent reviewing our manuscript. We appreciate the valuable feedback and the opportunity to revise our paper on ‘Productivity of short-rotation poplar crops. A case study in the NE of Romania’. The suggestions offered have been very helpful, and we appreciate the constructive comments.
We have included the comments immediately after this short letter and responded to them individually, indicating exactly how we addressed each concern or problem and describing the changes we have made. All the authors have approved the revisions. The changes are marked with ‘Track changes’ in the paper, and the revised manuscript is attached.
We hope the changes we made have resulted in a significantly improved manuscript that will better suit Forests journal. Thank you once again for the helpful input!
Sincerely,
Ciprian Palaghianu
June 23rd 2022
Here is a point-by-point response to the reviewers’ comments and concerns.
Reviewer 1
Comment 1: The introduction is tedious and the objectives are not clearly defined.
Response: In response to your comments on the introduction, we have revised it, adding more clarifying info.
Comment 2: Line 106 and 108: I don't detect the difference between both questions, I should clarify it more.
Response: We agree and we have updated the text to underline the difference. The first research question focuses on the genotype. The two clones' production differences are analysed, considering the crops with only one density (1667 trees · ha-1, the most common planting density). The second one is focused on planting density, analysing the differences in biomass accumulations of hybrid poplar crops at different planting densities.
Comment 3: Line 173: B is the dry weight of the biomass? should clear it up
Response: We add a clarifying note: B is the dry weight of the biomass.
Comment 4: Line 197-198: The statistical analysis is not clear. It must indicate which variables were studied and with respect to which factors. It must indicate which variables followed normal distributions and equality of variances and which did not.
Response: We thank the reviewer for pointing this out. We have revised the paragraph.
All the data sets have a normal distribution according to the Shapiro-Wilk test and the ANOVA test was applied to assess the significance of the biomass differences (Tukey test, for p≤0.05). Initially, there were several additional crops we want to select but they weren’t due to the lack of homogeneity considering the crop management (inconsistent maintenance activities and shorter rotation).
The annual biomass was analyzed with respect to the clone from a specific location. Then the tree parts biomass per hectare after five years was analyzed with respect to the clone from a particular site.
Comment 5: There are other factors besides the plantation density that influence the amount of biomass accumulated (Fertilization, Silvicultural treatments,...)
Response: In the manuscript's introduction, we acknowledged the influence of other factors such as fertilization or crop management (lines 51-58). However, in the case of our study, it was out of scope because we didn’t have robust data on these aspects. We already have a mix of establishment dates/planting densities/clones used which always makes the analysis and discussion lengthy and somewhat cumbersome.
Comment 6: Tables from other studies should not be reproduced in the discussion, the reference is sufficient.
Response: We thank the reviewer for pointing this out. We have removed the table from the manuscript, keeping the reference. However, to ensure easy access to these precise values, we have added them to the additional material.
Comment 6: The conclusions must be supported by the results and references must be avoided in them, as well as discussions.
Response: We thank the reviewer for pointing this out. We have removed the references and revised the whole section, shortening it and making it more concise.
Comment 7: Some references are difficult to access
Response: We have updated and added more identification information (we added DOI for some papers that did not specify it).

Reviewer 2 Report
The work presented here addresses important issues in terms of today's demand for woody biomass. What I miss in the introduction is information on the advantages of fast-growing poplar over e.g. energy willow, and examples of applications for biomass harvested from plantations in the context of a sustainable economy.
DOI:10.35812/CelluloseChemTechnol.2021.55.52
DOI:10.5604/01.3001.0015.3633
Conclusions should include the most important synthetic observations of the observations without restating the results. I would suggest reading this chapter and shortening it in parts. This information does not significantly affect the whole work.
Author Response
Dear Reviewer,
We thank you for the effort and the time spent reviewing our manuscript. We appreciate the valuable feedback and the opportunity to revise our paper on ‘Productivity of short-rotation poplar crops. A case study in the NE of Romania’. The suggestions offered have been very helpful, and we appreciate the constructive comments.
We have included the comments immediately after this short letter and responded to them individually, indicating exactly how we addressed each concern or problem and describing the changes we have made. All the authors have approved the revisions. The changes are marked with ‘Track changes’ in the paper, and the revised manuscript is attached.
We hope the changes we made have resulted in a significantly improved manuscript that will better suit Forests journal. Thank you once again for the helpful input!
Sincerely,
Ciprian Palaghianu
June 23rd 2022
Here is a point-by-point response to the reviewers’ comments and concerns.
Reviewer 2
Comment 1: The work presented here addresses important issues in terms of today's demand for woody biomass. What I miss in the introduction is information on the advantages of fast-growing poplar over e.g. energy willow, and examples of applications for biomass harvested from plantations in the context of a sustainable economy.
Response: In response to your comments on the introduction, we have revised it, adding more clarifying info and some specifically requested data (e.g. comparison info on poplar and willow in Romania, examples of applications for biomass). We have also used the two articles you mentioned and integrated that info.
Comment 2: Conclusions should include the most important synthetic observations of the observations without restating the results. I would suggest reading this chapter and shortening it in parts. This information does not significantly affect the whole work.
Response: We thank the reviewer for pointing this out. We revised the whole section, shortening it and making it more concise.

Reviewer 3 Report
My reading of this paper was a little challenging. Some of the "challenge" had to do with minor differences in terminology (e.g. "rods" vs. "sticks"; "humidity" vs "percent moisture:, others, but all are clear in context, so no real editing needed there. Another factor was the authors having to analyze and discuss 10 "crops" (presumably stands or plantations), they have a mix of establishment dates/planting densities/clones used. Those situations always make analysis and discussion lengthy and somewhat cumbersome, but what they have had to work with, they have done an acceptable job of discussion and interpreting and the paper should be published.
The paper will be of interest to all who plant or promote SRWC poplar as all want to know what others have observed in terms of biomass accumulation and response to management or edaphic factors. Despite the 10 sites having been termed "homogeneous", they acknowledge and nicely look at precipitation and they admit there could be unknown biotic factors and even things like hail that had specific impacts on specific stands.
A couple of minor suggested edits include:
Lines 62-65. The authors note that higher planting density reduces weed/grass competition "reducing evapotranspiration and water consumption". I assume the reduction is for the competition and not the stand as a whole. They should specify.
Beginning line 89. The authors state that SRWCs in Europe are 50,000 ha and then they state that hybrid poplar cover in Romania exceeds 73,000 ha. Is the fact that hybrid poplar coverage in Romania is much more than Europe as a whole ever through Romania to most is considered part of Europe. Is that because in Romania, poplar production is more for conventional forestry than SRWCs? They should explain.
Line 218. I have no idea what this sentence means. I have tried to derive the 7.71 Mg/ha myself just to see what it means, but I have been unable to do so. Help!
Author Response
Dear Reviewer,
We thank you for the effort and the time spent reviewing our manuscript. We appreciate the valuable feedback and the opportunity to revise our paper on ‘Productivity of short-rotation poplar crops. A case study in the NE of Romania’. The suggestions offered have been very helpful, and we appreciate the constructive comments.
We have included the comments immediately after this short letter and responded to them individually, indicating exactly how we addressed each concern or problem and describing the changes we have made. All the authors have approved the revisions. The changes are marked with ‘Track changes’ in the paper, and the revised manuscript is attached.
We hope the changes we made have resulted in a significantly improved manuscript that will better suit Forests journal. Thank you once again for the helpful input!
Sincerely,
Ciprian Palaghianu
June 23rd 2022
Here is a point-by-point response to the reviewers’ comments and concerns.
Reviewer 3
Comment 1: Lines 62-65. The authors note that higher planting density reduces weed/grass competition "reducing evapotranspiration and water consumption". I assume the reduction is for the competition and not the stand as a whole. They should specify.
Response: We thank the reviewer for pointing this out. We revised that paragraph. We guess the author of that study referred to the reduction of the potential water consumption (potential reduction due to a smaller number of water-consuming weeds), but logically, a higher planting density cannot lead to lesser water consumption.
Comment 2: Beginning line 89. The authors state that SRWCs in Europe are 50,000 ha and then they state that hybrid poplar cover in Romania exceeds 73,000 ha. Is the fact that hybrid poplar coverage in Romania is much more than Europe as a whole ever through Romania to most is considered part of Europe. Is that because in Romania, poplar production is more for conventional forestry than SRWCs? They should explain.
Response: We apologize for not being more explicit. The hybrid poplar cover in Romania is 73,000 ha but not all that area is managed as SRWC plantations (most of the stands are using long production cycles and consequently are not considered SRWCs). The area covered by hybrid poplar and managed as SRWC is only 2600 hectares. We have revised that paragraph, adding more info.
Comment 3: Line 218. I have no idea what this sentence means. I have tried to derive the 7.71 Mg/ha myself just to see what it means, but I have been unable to do so. Help!
Response: We tried not to complicate things further with additional explanations, but we have the advantage of knowing our manuscript better (which, in the end, is not to our advantage). We estimated the annual mean increase in biomass to be 7.71 Mg·ha-1 because considering the total biomass of all three crops (plantations) from Table 2 (the last line for the year 2015 is the total biomass accumulations), we obtained 45.86 + 38.27+31.53 =115.66 Mg·ha-1. To get the annual mean increase, we have to divide the outcome by 5 (years) and then by 3 (3 plantations). And we get that value. If you think it is necessary, we can add an explanatory paragraph.

Round 2
Reviewer 1 Report
The work has improved, but in my opinion there are aspects that should improve. Statistical analysis does not seem to be the most appropriate since, despite indicating normal data distributions, it does not refer to equality of variances, so parametric analysis is not appropriate.
The results are not presented according to what is indicated in the methods. There are other variables besides planting density that influence productivity that have not been mentioned and should be taken into account when extracting results,.....
Author Response
Dear Reviewer,
We thank you once again for the effort and the time spent reviewing our manuscript. We appreciate the valuable feedback and the opportunity to revise our paper on ‘Productivity of short-rotation poplar crops. A case study in the NE of Romania’. Your suggestions have been supportive, and we appreciate the constructive comments.
We have included the comments immediately after this short letter and responded to them individually, indicating exactly how we addressed each concern and describing the changes we have made. All the authors have approved the revisions. The changes are marked with ‘Track changes’ in the paper, and the revised manuscript is attached.
We hope our adjustments have resulted in a significantly improved manuscript that suits Forests journal better. Thank you again for the helpful input!
Sincerely,
Ciprian Palaghianu
July 3rd 2022
Comment 1: The work has improved, but in my opinion there are aspects that should improve. Statistical analysis does not seem to be the most appropriate since, despite indicating normal data distributions, it does not refer to equality of variances, so parametric analysis is not appropriate.
Response 1: We thank the reviewer for pointing this out. We have gone through the entire data analysis once again and we revised accordingly the data analysis section. We also had small doubts at first about using ANOVA, but we rely on the fact that it is a robust analysis. However, our hesitations have been confirmed, and we thank you for this observation. Thus, to eliminate any doubts related to the statistic investigation, we consider it better to use a non-parametric approach. Even if the initial ANOVA results are confirmed (the meanings of the differences between the means have not changed), it is a more appropriate method of data analysis in the context of data homogeneity. So, we made some changes. The influence of the genotype on biomass accumulation was tested using the non-parametric Kruskal-Wallis test. The annual biomass was analysed with respect to the clone from a specific location. Then biomass per hectare of the tree parts (after five growing seasons) was analysed with respect to the clone from a particular site. We decided to use a non-parametric test even though the tested distributions were normal, according to Shapiro–Wilk test. However, the condition of homogeneity of variance was not met, according to Levene’s test. Bonferroni-Dunn's multiple comparisons post hoc analysis (Zar, 2010) was used to assess the significance of the means differences.
Comment 2 The results are not presented according to what is indicated in the methods. There are other variables besides planting density that influence productivity that have not been mentioned and should be taken into account when extracting results,.....
Response 2:
Thank you for your observations. You have raised an important point here about other factors influencing biomass accumulations. In the manuscript's introduction, we acknowledged the influence of such factors (lines 51-58). The aspect is exciting, and we would like to study it in the following research, but using a new experimental specifically designed to allow further analysis. However, in the case of our study, we didn't have robust data on these aspects.
In conclusion, we admit the influence of other factors, but the experimental design and the field records did not allow other factors isolation or analysis. Since the crops are homogeneous in terms of climatic conditions, soil type, installation method and treatment, the differences in biomass production between the plantations (crops) with different densities could be explained by different weather conditions (primarily the rainfall in May-June) or biotic disturbances. Minor changes were made in the final section.
